# Identification of anisomycin, prodigiosin and obatoclax as compounds with broad-spectrum anti-parasitic activity

Gretchen Ehrenkaufer[1], Pengyang Li[2], Erin E. Stebbins[3], Monica M. Kangussu-Marcolino[1], Anjan Debnath[4], Corin V. White[5], Matthew S. Moser[5], Joseph DeRisi[5], Jolyn Gisselberg[6], Ellen Yeh[6,7,8], Steven C. Wang[4], Ana Hervella Company[4], Ludovica Monti[4], Conor R. Caffrey[4], Christopher D. Huston[3], Bo Wang[2,9], Upinder Singh[1,7]*

1 Division of Infectious Diseases, Department of Internal Medicine, Stanford University, Stanford, CA, United States of America, 2 Department of Bioengineering, Stanford University, Stanford, CA, United States of America, 3 Department of Medicine, University of Vermont Larner College of Medicine, Burlington, Vermont, United States of America, 4 Center for Discovery and Innovation in Parasitic Diseases, Skaggs School of Pharmacy and Pharmaceutical Sciences, University of California, San Diego, La Jolla, CA, United States of America, 5 Department of Biochemistry and Biophysics, University of California, San Francisco, San Francisco, California, United States of America, 6 Department of Biochemistry, Stanford Medical School, Stanford University, Stanford, CA, United States of America, 7 Department of Microbiology and Immunology, Stanford University, Stanford, CA, United States of America, 8 Department of Pathology, Stanford University, Stanford, CA, United States of America, 9 Department of Developmental Biology, Stanford University School of Medicine, Stanford, CA, United States of America

* usingh@stanford.edu

**Data Availability Statement:** All relevant data are within the manuscript and its Supporting Information files.

## Abstract

Parasitic infections are a major source of human suffering, mortality, and economic loss, but drug development for these diseases has been stymied by the significant expense involved in bringing a drug though clinical trials and to market. Identification of single compounds active against multiple parasitic pathogens could improve the economic incentives for drug development as well as simplifying treatment regimens. We recently performed a screen of repurposed compounds against the protozoan parasite *Entamoeba histolytica*, causative agent of amebic dysentery, and identified four compounds (anisomycin, prodigiosin, obato-clax and nithiamide) with low micromolar potency and drug-like properties. Here, we extend our investigation of these drugs. We assayed the speed of killing of *E. histolytica* trophozo-ites and found that all four have more rapid action than the current drug of choice, metroni-dazole. We further established a multi-institute collaboration to determine whether these compounds may have efficacy against other parasites and opportunistic pathogens. We found that anisomycin, prodigiosin and obatoclax all have broad-spectrum antiparasitic activity *in vitro*, including activity against schistosomes, *T. brucei*, and apicomplexan para-sites. In several cases, the drugs were found to have significant improvements over existing drugs. For instance, both obatoclax and prodigiosin were more efficacious at inhibiting the juvenile form of *Schistosoma* than the current standard of care, praziquantel. Additionally, low micromolar potencies were observed against pathogenic free-living amebae (*Naegleria fowleri*, *Balamuthia mandrillaris* and *Acanthamoeba castellanii)*, which cause CNS infection and for which there are currently no reliable treatments. These results, combined with the

**Funding:** Funding for this project was provided by SPARK Translational Research Program at Stanford University; funding from the Stanford Department of Medicine Translational Research and applied Medicine Center; The Child Health Research Institute, Lucile Packard Foundation for Children's Health; and through the NIH (R21-AI123594 to US). Giardia, Naegleria and Acanthamoeba work was supported by 1KL2TR001444, R21AI133394 and R21AI141210 to AD. Cryptosporidium work was supported by grants from The Bill and Melinda Gates Foundation (OPP1132796) and the NIH (R21-AI130807) to CDH. PL and BW were supported by a Beckman Young Investigator Award. Screening of T. brucei was supported by NIH-NIAID R21AI133394 and R21AI141210. The funders had no role in study design, data collection and analysis, decision to publish, or preparation of the manuscript.

**Competing interests:** The authors have declared that no competing interests exist.

previous human use of three of these drugs (obatoclax, anisomycin and nithiamide), support the idea that these compounds could serve as the basis for the development of broad-spectrum anti-parasitic drugs.

## Author summary

Parasitic diseases are a major cause of human morbidity and mortality worldwide, as well as a significant economic drain in developing countries. Many parasites have limited treatment options with low efficacy and significant side effects, however research into new therapeutics suffers from a lack of investment. In this study, we characterize four potential anti-parasitic drugs: anisomycin, nithiamide, prodigiosin and obatoclax. These drugs were previously shown to effectively inhibit *Entamoeba histolytica*, the parasite that causes amebic dysentery. Here, we demonstrate that these drugs have activity against a wide variety of parasites from different taxonomic groups. Additionally, we assessed the speed of killing of these compounds against *E. histolytica* and the brain pathogen *Balamuthia mandrillaris*, and show that several are faster acting than current drugs. Two of these drugs (prodigiosin and obatoclax) had broad-spectrum activity, including against life stages not treated by current drugs such as juvenile schistosome worms, and three (obatoclax, nithiamide and anisomycin) have been used previously in humans. Although more study will be needed to adapt these drugs to the varying requirements for treatment of each parasitic disease, this work is a promising beginning towards identifying drugs against multiple parasites that are human pathogens.

## Introduction

Parasitic diseases cause a significant public health burden, especially in the developing world. A 2013 survey of the causes of mortality worldwide estimated ~1 million deaths due to parasitic diseases, with parasitic protists such as *Plasmodium* being the most common [1]. In addition to this loss of life, significant morbidities such as cognitive impairment and growth stunting often result from parasitic infections [2]. Despite this large impact on human health, drug discovery efforts to develop new treatments for parasitic diseases have significant under-investment. Given the high cost of bringing a drug to market [3], economic considerations are challenging for developing therapies for diseases mostly prevalent in low resource environments. At the same time, biological barriers also exist which hinder successful drug development. These include the tendency of many parasites to become resistant to treatment as well as drug toxicity, which can be severe with drugs for eukaryotic pathogens due to conservation between parasite and host pathways [4]. In addition, many parasites have multiple life stages, which may have differing drug susceptibilities. Due to these issues, the idea of repurposing drugs originally developed for other diseases to treat parasitic infections has been growing in popularity. This approach can significantly lower the cost of bringing drugs to market by reducing the need for extensive pre-clinical testing and clinical trials [5].

We recently performed a screen of repurposing libraries, totaling ~4000 compounds, to identify compounds targeting the protozoan parasite *Entamoeba histolytica* [6]. From this work we identified four compounds: nithiamide (a nitroimidazole agent), anisomycin (an antibiotic isolated from *Streptomyces*), prodigiosin (a natural pigment isolated from a bacterium), and obatoclax (a synthetic analog of prodigiosin thought to inhibit BCL-2). Three of

these, prodigiosin, obatoclax and anisomycin, had activity against both the trophozoite and cyst stages of *E. histolytica* and, importantly, were also active against metronidazole-resistant parasites. Of these compounds, anisomycin, nithiamide and obatoclax all have been used in humans, either in clinical trials or historically [7–9]. This finding validates our approach of screening with repurposed drug libraries, as drugs that have previously been used safely in humans should have a faster and cheaper regulatory route.

We next asked whether these four compounds are active against a broad-range of parasitic pathogens including (i) another anaerobic enteric parasite (*Giardia lamblia*), (ii) free-living amebae (*Naegleria fowleri*, *Acanthamoeba castellanii*, and *Balamuthia mandrillaris*), (iii) apicomplexan parasites (*Plasmodium falciparum* and *Cryptosporidium parvum*), (iv) trypanosomatids (*Trypanosoma brucei*), and (v) multi-cellular worm (*Schistosoma mansoni*). *Giardia lamblia* is an anaerobic protozoan parasite that inhabits the small intestine of humans and other animals. It is the most common intestinal parasite [10], causing severe diarrhea in over 100 million people per year [11]. The free-living amebae *Naegleria fowleri*, *Acanthamoeba castellanii* and *Balamuthia mandrillaris* are opportunistic pathogens that can cause rare but dangerous infections of the central nervous system (CNS); *Acanthamoeba* can cause ocular disease and both *Balamuthia* and *Acanthamoeba* can cause systemic disease. Current drug regimens for CNS infections with these amebae are sorely inadequate, and even with treatment fatality rates are >90% [12]. Parasites of the phylum Apicomplexa are responsible for many of the most common parasitic diseases, including malaria. *P. falciparum* is responsible for ~80% of malaria cases worldwide [13], causing severe fever due to the lysis of infected red blood cells, and leading to an estimated 620,000 deaths per year [1]. *Cryptosporidium parvum* has been recognized as a serious cause of childhood diarrhea in the developing world [14]. Human African Trypanosomiasis (HAT), which is caused by two subspecies of *T. brucei*, is lethal in humans if not treated appropriately. There is no vaccine and treatment relies on a small number of mostly old drugs that must be administered parenterally, have a range of often serious side-effects and for which resistance has been a problem [15–17]. Schistosomiasis, caused by trematode worms of the genus *Schistosoma*, affects an estimated 250 million people in at least 76 countries worldwide [18]. Symptoms include fever and bloody diarrhea, as well as hepatic and renal pathology, caused by the large numbers of eggs laid by female worms [18].

We now further characterize anisomycin, prodigiosin, obatoclax, and nithiamide for activity against a broad range of parasites and show that all compounds have activity against at least one other parasite group. Prodigiosin and its related compound obatoclax had the broadest-spectrum activity, showing the ability to kill at least one parasite from each of the groups listed above. Importantly, both prodigiosin and obatoclax effectively kill juvenile schistosome worms with high efficacy. We performed a preliminary investigation of *in vivo* activity using a mouse model of *Cryptosporidium* infection but found no significant improvement in parasite burden. Despite this negative result, important information about the tolerability and dosage of oral prodigiosin was gained. Anisomycin was also a promising lead, due to its activity against *Naegleria*, apicomplexan parasites, and *T. brucei*, along with its previous history of established human use. This study provides exciting new leads for drug development efforts for single compounds to target multiple parasites that cause serious human disease.

## Results

### *E. histolytica* speed of killing

Rapid action of antiparasitic drugs is important for improving efficacy and reducing treatment duration. Additionally, better understanding of the kinetics of activity can offer insight into the mechanism of action of lead compounds [19]. In order to determine speed of killing of *E.*

*histolytica* trophozoites, we performed a time course experiment in which parasite growth was assayed at 10, 24 and 48h post treatment. The four compounds were assayed at 2 times the previously established $EC_{50}$ [6] and compared to metronidazole and auranofin [20], also at 2x the $EC_{50}$. Fluorescence after incubation with FDA was compared to parasites treated with 0.6% DMSO from the same time point and three independent biological replicates were performed. All four compounds inhibited trophozoite growth more rapidly than metronidazole (**Fig 1**). Prodigiosin and nithiamide were the fastest acting, with ~50% inhibition by as early as 10h post treatment. The result with nithiamide was especially notable, considering that it is chemically similar to metronidazole and they are presumed to have the same mechanism of action. Anisomycin and obatoclax were slower acting, but both had strong activity by 24h, compared to metronidazole which did not have significant inhibition until 48h of treatment. It is important to note that control parasites did not exhibit significant growth over this short incubation period (compared to parasite number at time zero), indicating that there was likely an actual reduction in parasite viability, and not simply an inhibition of growth.

## Parasite drug susceptibility testing

In order to determine if the compounds with activity against *Entamoeba* are also active against other parasitic diseases, we tested their activity against (i) other anaerobic enteric parasites (*Giardia lamblia*), (ii) free-living amebae (*Naegleria fowleri*, *Acanthamoeba castellanii*, and *Balamuthia mandrillaris*), (iii) apicomplexan parasites (*Plasmodium falciparum* and *Cryptosporidium parvum*), (iv) trypanosomatid (*Trypanosoma brucei*), and (v) multi-cellular worm (*Schistosoma mansoni*). The $EC_{50}$s for all drugs and parasites tested are shown in **Table 1**. A survey of publicly available toxicity and pharmicokinetic data is shown in **Table 2**.

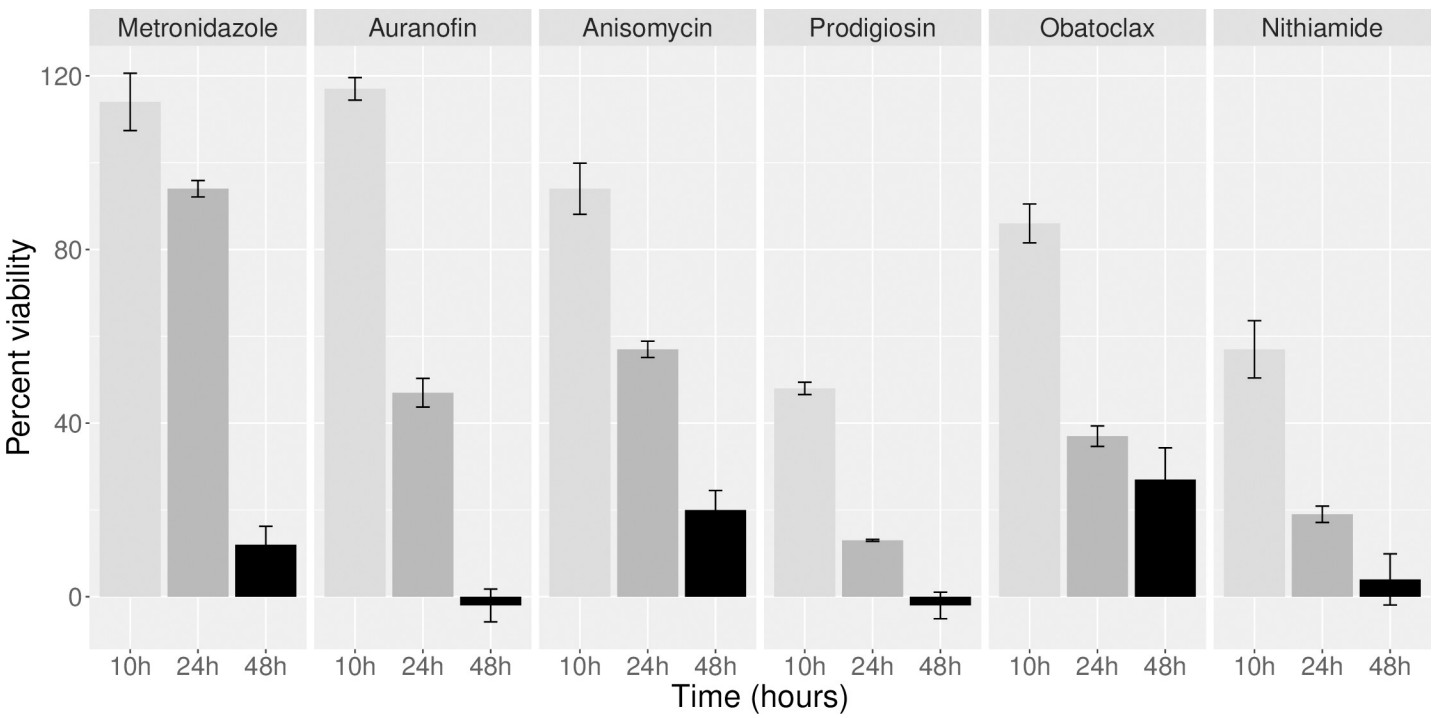

**Fig 1. Speed of killing for *Entamoeba histolytica*.** Results from experiment to assess kinetics of drug action against *E. histolytica* trophozoites. Drugs were assayed at the following concentrations (2x the previously measured $EC_{50}$ [6]): metronidazole, 17µM; auranofin 0.5µM; anisomycin, 1.4µM; prodigiosin, 1.4µM; obatoclax, 1µM; nithiamide, 10µM. Graph shows FDA signal as a percent of DMSO control. Killing was assayed at 10, 24 and 48 hours, and three biological replicates were performed for each data point.

**Table 1. Potency of candidate compounds in various parasite systems.**

| Parasite | Anisomycin | Prodigiosin | Obatoclax | Nithiamide | Control |
|---|---|---|---|---|---|
| *Entamoeba* | 0.7 | 0.7 | 0.5 | 5 | Metronidazole (8.9μM) |
| *Giardia* | 18.5 | 3.8 | 0.9 | 2.6 | Metronidazole (6.4μM) |
| *Naegleria* | 4.7 | 6.4 | 3.6 | no activity | Amphotericin (0.2μM) |
| *Acanthamoeba* | no activity | 2.2 | 0.5 | no activity | PHMB (9.8μM) |
| *Balamuthia* (trophozoites) | no activity | 4 | 1.5 | no activity | Nitroxoline (2.8μM) |
| *Balamuthia* (cysts) | no activity | 3.8 | 1.7 | no activity | Nitroxoline (15.5 μM) |
| *Plasmodium* | 0.1 | nt* | nt* | no activity | Atovaquone (23nM) |
| *Cryptosporidium* | 0.08 | 0.09 | 2.1 | no activity | Nitazoxanide (2.07μM) |
| *Schistosoma* (juvenile) | no activity | 1 | 0.6 | no activity | Praziquantel (4.7μM) |
| *T. brucei* | 0.10 | 0.03 | 0.04 | 3.6 | Pentamidine (0.11μM) |

The $EC_{50}$ values for anisomycin, prodigiosin, obatoclax and nithiamide for each parasite tested are shown. All concentrations are micromolar. Results for control compounds are from the same experiment except as indicated: metronidazole (*Entamoeba*) [6], nitroxoline (*Balamuthia*) [21], Atovaquone (*Plasmodium*) [22]; praziquantel (juvenile *Schistosoma*) [23]. * indicates that compound was not tested as it was incompatible with the assay.

**Giardia lamblia.** Current treatment of *Giardia* is based on nitroimidazole agents (tinidazole and metronidazole) and nitazoxanide, a thiazolide-family compound. However, treatment failure is noted in up to 20% of isolates and re-treatment or combination therapies are needed [40]. In our testing, we found that *Giardia* was susceptible to nithiamide, as previously reported [35], with an $EC_{50}$ of 2.6μM. This compound belongs to the nitroimidazole class, similar to metronidazole, although it was slightly more efficacious (metronidazole $EC_{50}$ = 6.4μM). Additionally, we found that both prodigiosin and obatoclax inhibited *Giardia* with good potency ($EC_{50}$s 3.8 and 0.9μM respectively) (**Table 1**). As these compounds are chemically

**Table 2. Pharmacokinetic and cytotoxicity properties.**

| Drug | Anisomycin | Prodigiosin | Obatoclax | Nithiamide |
|---|---|---|---|---|
| *in vivo* data | | | | |
| Animal | rat | mouse | mouse | mouse |
| Toxicity | $LD_{50}$: 72mg/kg (oral) [1] | $LD_{50}$: 18mg/kg (ip) [2] | >5mg/kg (IV) [3] | $LD_{50}$: 300mg/kg (ip) [4] |
| Cmax in μM | 0.01 (150mg/kg subQ) [5] | na | 0.24 (4.8mg/kg IV) [6] | na |
| Cytotox ($CC_{50}$) | | | | |
| HepG2 | 0.39μM | 0.21μM | 0.21μM | no activity |
| HEK293 | 0.08μM | 0.13μM | 0.8μM | no activity |
| MDCK | na | no activity [7] | 1.4μM [8] | na |
| PBMCs | na | 241μM [9] | >4μM [3] | na |
| Vero | 45μM [10] | na | 6.6μM [11] | na |
| CHO | na | na | na | no activity [12] |
| Huh-7 | 2.4μM [13] | na | 0.38μM [14] | na |
| A549 | no activity [15] | na | 9.9μM [16] | na |
| hFF-1 | >25μM | 0.14μM | 0.11μM | no activity |

$LD_{50}$ and Cmax for published animal studies, and a selection of publicly available cytotoxicity data for each compound are shown. Route of administration and dosage in parenthesis. 'na' indicates that no data was available. Results for HepG2 and HEK293 cytotoxicity from Calibr (reframedb.org). Cytotoxicity against human foreskin fibroblast cells (hFF-1) was determined as mentioned in methods. Other data are as follows: 1. [24]; 2. [25]; 3. [26] Truedel 2007; 4. [27]; 5. [28]; 6. [29]; 7. [30]; 8. [31]; 9. [32]; 10. [33]; 11. [34]; 12. [35]; 13. [36] (pubchem AID:449705); 14. [37] (pubchem AID:742238); 15. [38] (pubchem AID:720523); 16. [39]

unrelated to metronidazole, and were effective against metronidazole resistant *Entamoeba* [6], they may make good candidates for treatment of resistant strains of *Giardia* [41]. Some anti-*Giardia* activity was also seen with anisomycin, but efficacy was poor ($EC_{50}$ of 18.5µM) compared to the other compounds tested and to current treatments.

**Free-living amebae *Naegleria*, *Acanthamoeba* and *Balamuthia*.** Current treatment of CNS disease caused by the free-living ameba is based on a multi drug regimen often including antibiotics such as azithromycin and pentamidine as well as amphotericin B and miltefosine. However, these protocols have poor efficacy and mortality rates remain very high [42]. We found that all three amebae were inhibited by prodigiosin and obatoclax, with $EC_{50}$s in the low micromolar range (**Table 1**). Additionally, anisomycin was active against *Naegleria*. These results compare favorably to current treatments; for instance, miltefosine killing of *Naegleria* has an $EC_{50}$ of ~48µM [43]. However, it is important to note that *in vitro* $EC_{50}$s do not necessarily translate to *in vivo* success. For instance, amphotericin has sub-micromolar $EC_{50}$ *in vitro* but is not effective in patients. An additional aspect in the treatment of these parasites is that they can cause infection in multiple different tissues (CNS, eye, skin), requiring distinct properties such as CNS penetrance. Alternate formulations (such as for ocular topical application) of medicinal chemistry efforts to enhance CNS availability may be important for developing these compounds into useful drugs. Significantly, both compounds also were active against *Balamuthia* cysts, which are typically less sensitive to many drugs than are *Balamuthia* trophozoites [21].

**Balamuthia phenotypic assays.** Given the potency of obatoclax against both life stages of *Balamuthia*, as well as evidence that it can penetrate the CNS [29], we wanted to further characterize the activity of this drug with a variety of phenotypic assays (**Fig 2A**). To determine the kinetics of the killing of *Balamuthia* trophozoites, we performed a similar assay to that used for *Entamoeba*. The compound nitroxoline, which has previously been shown to have rapid *in vitro* activity [21] was used as a control. Both compounds showed significant reduction of signal compared to the DMSO control by 8 hours and had >90% inhibition by 24 hours after treatment (**Fig 2B**), indicating rapid action that could be critical in the treatment of this fatal disease.

It has previously been found that some drugs can induce encystation in *B. mandrillaris* [21]. This property could slow disease progression, as cysts are non-proliferative, though it could also make clearance of the infection more difficult. To determine whether obatoclax has this property, we treated *Balamuthia* trophozoites with obatoclax and nitroxoline, both at 2x the $EC_{50}$ concentration for 72h. We determined that, in contrast to nitroxoline, obatoclax did not significantly enhance encystation, compared to the DMSO control (**Fig 2C**).

Finally, we explored the potential of obatoclax to delay *Balamuthia* recrudescence after treatment. For this assay, trophozoites were treated with drug at 2x the $EC_{50}$ for 72h, then washed and seeded on a monolayer of hFF cells in 24-well plates. Cells were monitored at 3, 5, 7 and 9 days, and the percent of intact monolayer was recorded. Since untreated or DMSO treated *Balamuthia* trophozoites quickly destroy host cells, this assay gives a clear measure of viability after drug removal. We found that the obatoclax treated trophozoites were able to completely destroy the hFF monolayer in 3 days, showing that 72h treatment was insufficient to prevent recrudescence (**Fig 2D**). This finding may indicate that more extended treatment with obatoclax would be required in a clinical setting.

**Schistosoma mansoni.** The current drug of choice, praziquantel, is cheap and effective against adult schistosomes, but has reduced activity in killing juveniles [45]. Additionally, resistance to the drug has been seen both *in vitro* and in the clinic [46]. The recovery of juvenile worms after praziquantel treatment may contribute to drug resistance in human populations; for these reasons, attempts to identify drugs that are effective at all life stages is a vital

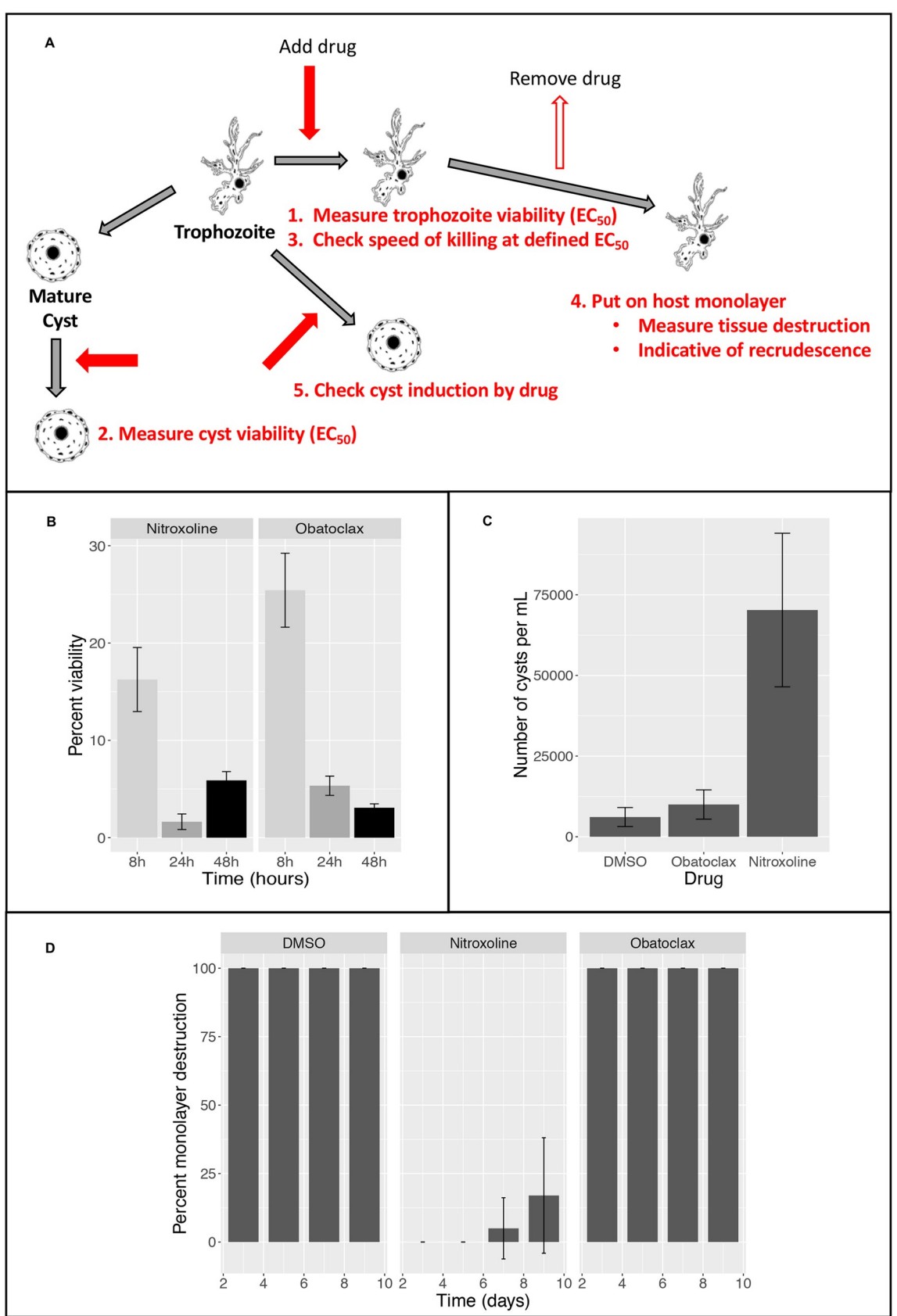

**Fig 2. Phenotypic assays for *Balamuthia mandrillaris*. (A)** Assays performed: Schematic representing the workflow for each of the assays performed for *Balamuthia*. Red arrows represent addition of drug and outline arrows represent drug removal. **(B)** Speed of killing: Results from experiment to asses kinetics of drug action against *Balamuthia* trophozoites. Drugs were assayed at 2x the EC$_{50}$ as previously assessed [21, 44]: nitroxoline at 15.6μM and obatoclax at 3.2μM. Graph shows FDA signal as a percent of 0.5% DMSO control. Killing was assayed at 8, 24 and 48 hours, and three biological replicates were performed for each data point. **(C)** Cyst induction: Trophozoites were treated for 72h with obatoclax (3.2μM), nitroxoline (15.6μM) or DMSO (0.5%), in 24-well plates, then the numbers of trophozoites and cysts were counted. Nitroxoline induces substantial cyst formation, as previously observed [21], but obatoclax is similar to control. **(D)** Parasite recrudescence: Trophozoites were treated for 72h with obatoclax (3.2μM), nitroxoline (15.6μM) or DMSO (0.5%), in flasks, then drug was removed, and *Balamuthia* were seeded on a monolayer of mammalian cells. The monolayer was rapidly destroyed by trophozoites after obatoclax treatment, indicating that this concentration and/or duration of treatment was insufficient to prevent recrudescence.

area of research [46, 47]. We tested our four lead compounds in a viability assay against juvenile forms of *Schistosoma mansoni*. Of the compounds we tested, only prodigiosin and obatoclax were effective, with EC$_{50}$ values of 1μM and 0.6μM respectively (**Table 1**). To determine whether the worms could recover from the drugs, we transferred the worms after 72h 5μM drug treatment to fresh media and monitored activity. For both prodigiosin and obatoclax, worm motility was not restored even after 72 hours recovery, contrasting significantly with results seen after praziquantel treatment where juvenile worms were found to have significantly better recovery after drug treatment than adult worms [48, 49]. Imaging of prodigiosin and obatoclax treated worms revealed that both drugs resulted in significant damage to the tegument, with numerous areas of blebbing and separation from the worm body (**Fig 3**), a phenotype previously seen in with other anti-schistosomal drugs, and associated with increased ion flow across the tegument, exposure to host immune system, and eventual loss of viability [50, 51].

***Trypanosoma brucei.*** Treatment for Human African Trypanosomiasis, caused by the parasite *T. brucei*, varies based on subspecies and disease stage but includes intravenous dosing of drugs such as melarsoprol which can have significant side effects [16]. Fexinidazole has been recently approved for treatment of gambiense HAT [52] and clinical trials are ongoing for its utility against rhodesiense HAT. All four compounds were active against *T. brucei* with EC$_{50}$ values ranging from 3.6 μM (nithiamide) to 0.03μM (prodigiosin) (**Table 1**). Except for nithiamide, all were more potent than the control compound, pentamidine, in this assay.

**Apicomplexan parasites.** The first line treatment for uncomplicated *P. falciparum* malaria involves treating malaria with artemisinin-based combination therapy [13], involving combining treatments such as such as artemether + lumefantrine or artesunate + amodiaquine. These treatments can be highly effective in most cases; however drug resistance is a major problem, already affecting even advanced treatments [53]. Hence, new drugs for this parasite are constantly needed. Due to the fluorescence-based nature of the assay used we were unable determine efficacy of either prodigiosin or obatoclax, both of which had interfering fluorescence. However, there are previously published reports of anti-malarial activity with this compound class [54]. We did find potent activity with anisomycin (EC$_{50}$ 0.1μM, **Table 1**). This is consistent with previously published results [55], indicating a potential for this drug to be added to the anti-malarial arsenal. Nithiamide had no activity in this parasite.

*C. parvum* is an intestinal parasite that causes diarrheal symptoms and can result in malnutrition and impaired growth in children [56]. The most important current treatment is nitazoxanide [57], which can reduce diarrhea symptoms in adults but has reduced efficacy in children and immunocompromised patients. Of our compounds, anisomycin, prodigiosin and obatoclax all had potent activity (EC$_{50}$s: 0.08μM, 0.09μM and 2.1μM respectively) (**Table 1**). The efficacy of anisomycin against both apicomplexan parasites was intriguing, opening the possibility that it could have wider use against other members of this group such as *Toxoplasma* and *Cyclospora*. Although we could not assay prodigiosin or obatoclax killing

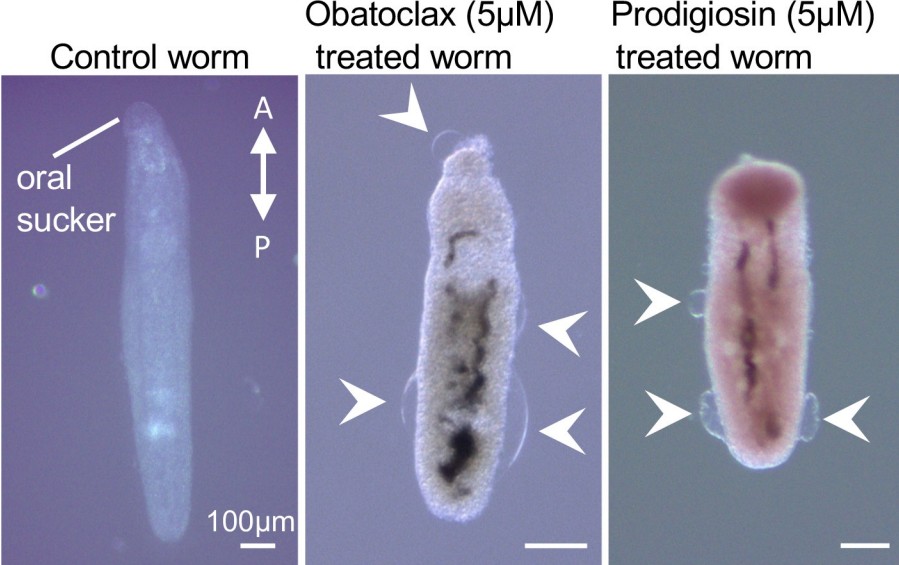

**Fig 3. *S. mansoni* phenotypes after treatment with obatoclax and prodigiosin.** Imaging of juvenile *Schistosoma* after 72h treatment with DMSO control (panel 1) 5μM obatoclax (panel 2) or 5μM prodigiosin (panel 3). Note changes in gross morphology as well as tegument blebbing (arrows). Position of oral sucker and anterior-posterior axis are indicated.

of *Plasmodium*, their strong inhibition of *C. parvum* as well as many of the other parasites tested point to it as a potential lead compound for the development of a broad-spectrum anti-parasitic agent.

**_Cryptosporidium in vivo_ model for testing prodigiosin efficacy.**    Based on these promising results, we decided to test prodigiosin in an *in vivo* mouse model of *Cryptosporidium* infection. NOD SCID gamma mice were infected with *C. parvum* oocysts and the infection was allowed to progress for seven days. On day seven, mice were treated orally with either vehicle (5% DMSO in 1% HPMC), prodigiosin (25mg/kg), or MMV665917 (60mg/kg), a recently discovered lead compound with *in vivo* activity [58], as a positive control. Both compounds were dosed twice daily for four days, and oocyst shedding was monitored daily. Comparison of the number of oocysts shed on day 7 (before start of treatment) and day 11 showed that while MMV665917 significantly reduced the number of oocysts, oocyst shedding increased during this time period for mice treated with either prodigiosin or the vehicle control (**Fig 4**). In addition, some weight loss was observed in prodigiosin mice, although this effect reversed upon cessation of treatment. Reasons for this lack of *in vivo* efficacy despite *in vitro* potency include potential pharmacokinetic issues or insufficient dosing. Unfortunately, considering the side effects noted with prodigiosin at the dose used, increasing dosage is likely to be unsuccessful. Obatoclax, which has a record of safe human use [7] and also had *in vitro* activity against *C. parvum* (**Table 1**) is a potential candidate for further *in vivo* studies.

## Potential for BCL2-like inhibitors targeting parasitic pathogens

Obatoclax, which is active against all parasites assayed in our study, is a potent inhibitor of the apoptosis regulator BCL-2 [59]. However, except for *Schistosoma* all the organisms are single-celled protists which do not have apparent BCL-2 homologs. In order to gain a better understanding of the mechanism of action of obatoclax, we obtained a number of BCL-2 inhibitors (Venetoclax, Navitoclax, A-1331852, A-1210477 and S63845) and tested them in our assays

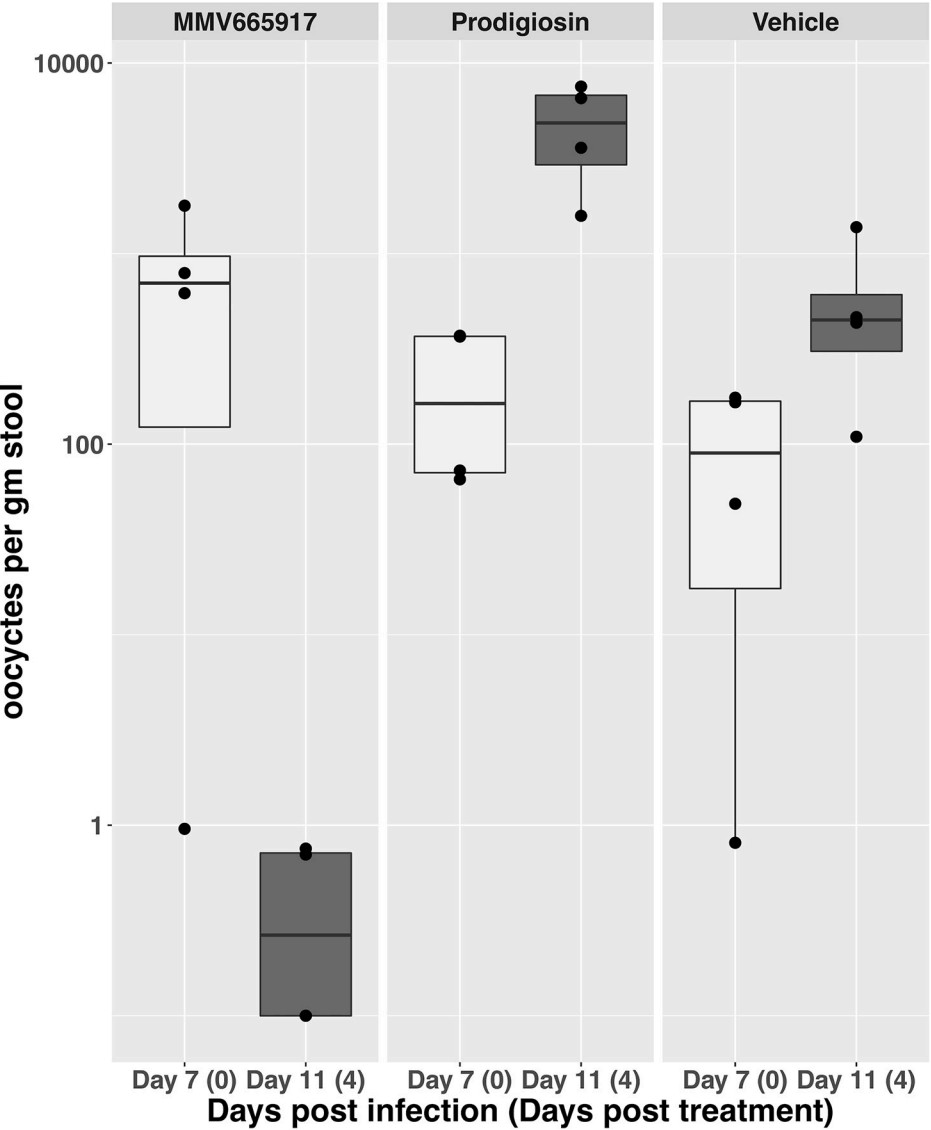

**Fig 4. Effect of prodigiosin on *C. parvum* infection in an *in vivo* mouse model.** Mice infected with *C. parvum* oocysts were treated beginning on day 7 post infection with Vehicle (DMSO), positive control (MMV665917 at 60 mg/kg twice daily) or prodigiosin at 25mg/kg twice daily. Number of oocysts in the stool were counted at day 7 (day one of treatment) and day 11 (day four of treatment). Four mice per condition were used. Results for each individual animal are shown by a black dot (•).

against a subset of the parasites. Surprisingly, no effect was seen on viability of juvenile schistosome worms, at concentrations up to 20μM (**S1 Table**), indicating that obatoclax may have another mechanism of action in schistosomes. This may be due to sequence divergence, as the *S. mansoni* BCL-2 (XP_018654288.1) is only ~20% identical and ~60% similar to the human protein based on clustal omega alignment. Of the protists, some weak activity was seen against *Naegleria* with A-1331852 (EC$_{50}$ ~30μM). Intriguingly, Navitoclax, A-1331852, A-1210477 and S63845 all had activities against *Cryptosporidium* in the 3–20μM range. No effect was seen on any of the other protists tested (*E. histolytica*, *B. mandrillaris*, *T. brucei*). A blastP search of the *C. parvum* genome indicates that the protein with the closest homology to human BCL-2 is an uncharacterized protein cgd4_3520 (e-value 0.026). Given the absence of an obvious

*Cryptosporidium* BCL-2 homologue, obatoclax likely effects *C. parvum* growth either by inhibiting host cell BCL-2 or via a non-BCL-2 dependent mechanism. Overall, these results indicate that obatoclax most likely kills the parasites tested though a non-BCL-2 dependent mechanism.

## Discussion

Development of novel therapeutics against parasitic pathogens would benefit from identification of drugs that can work against multiple organisms as a means to reduce drug development and regulatory costs. We previously identified four compounds with potent activity against the protozoan parasite *E. histolytica*, and we now expand our exploration of these compounds looking at their potential as broad spectrum anti-parasitic compounds. We found that two of the compounds, prodigiosin and obatoclax, have broad spectrum anti-parasitic activity, including against *T. brucei* and the juvenile form of *Schistosoma mansoni* [47]. Of these, obatoclax is particularly interesting, due to its lower cost and safe record of human use. Both compounds have demonstrated ability to kill tumor-derived cell lines, but lower toxicity towards non-malignant cells such as PBMCs and MDCK cells (**Table 2** and [30]). Anisomycin had efficacy in fewer systems, but its good potency against both apicomplexan parasites tested, and history of safe human use [8, 60] make it a potential candidate for further development.

In considering one drug for multiple pathogens, we have to also consider the Target Product Profile (TPP) for each disease. *Entamoeba*, *Giardia*, and *Cryptosporidium* all cause gastrointestinal disease. For these parasites, oral formulations that can kill the parasites in the intestinal lumen are ideal. In addition, treatments that are cheap and have a good safety profile are necessary, as they would likely be used widely in resource-poor endemic regions. Anisomycin has good efficacy against *Entamoeba* and *Cryptosporidium*, and significant animal toxicity and pharmacokinetic data including oral and IV treatment in rodents (**Table 2** and [24, 28]). Chronic treatment in female macaques was tolerated in doses up to 64mg/kg daily, for 30 weeks [24]. These data, along with its past use as an oral antibiotic [8, 60], supports the idea that anisomycin could, with further *in vivo* testing and clinical trials, become a useful addition to current treatment options. Prodigiosin was able to inhibit all three parasites *in vitro*. However, the toxicities noted during *in vivo* treatment may indicate potential difficulties in developing a safe drug for human use. Obatoclax had similar $EC_{50}$s and has a more favorable safety profile, making it a better candidate for further *in vivo* testing and development. In clinical trials for leukemia, IV treatment of obatoclax resulted in mostly transient neurological side effects (euphoria, dizziness, somnolence) [7], with a maximum tolerated dose of 28mg/m$^2$ [61]. However, it would need to be tested in oral formulation for use against these parasites. It is possible that the failure of prodigiosin in our *in vivo Cryptosporidium* trial was in part due to pharmacokinetic issues leading to low drug concentrations in the colon. Development of obatoclax or any drug for this use would have to take these properties into account; absorption rate in the small intestine and interactions with the colonic epithelium would have to be monitored in any new formulation.

Like the GI parasites, an effective treatment for schistosomes ideally should be cheap and have few side effects. However, ability of a compound to achieve significant plasma concentrations is important, as worms are found in the bloodstream. The current standard of care, praziquantel, is inexpensive and safe enough to be given prophylactically as a single oral dose in at-risk areas [62]. However, for treatment of infected individuals, the ineffectiveness against juvenile stages may require repeated dosing to achieve a cure. Thus, a compound with activity against the juvenile stages would be of interest. Obatoclax, which had potent activity against juvenile worms, could be used either as an IV treatment for severe cases or reformulated into an oral medication. Malaria has a similar TPP to schistosomiasis, requiring a safe, low cost

drug capable of achieving high levels in the blood stream. From our candidate drugs, anisomycin had good potency, and a promising pharmacokinetic profile [28].

In contrast to these parasites, the free-living amebae, *Naegleria*, *Balamuthia* and *Acanthamoeba*, rarely cause disease in humans, but when they infect the CNS are almost universally fatal with no good current therapies. Two different TPPs would need to be considered: one for CNS disease (with any of the three amebae) where patients are urgently ill and a second for skin or systemic symptoms (with *Acanthamoeba* or *Balamuthia*) where patients are chronically ill. Additionally, *Acanthamoeba* causes keratitis, a serious eye infection found most prevalently in contact lens wearers. Amebic keratitis is treatable, but current regiments are prolonged and have significant side-effects, largely due to the need to eliminate the resistant cyst form of *Acanthamoeba* [63]. To determine if prodigiosin or obatoclax could be a useful treatment for keratitis would require testing for effective killing of cysts, as well as re-formulation for ocular use. It should be noted that although activity against *Acanthamoeba* cysts was not tested, we did see efficacy of obatoclax against cysts of *Balamuthia*, a related parasite [64]. For the CNS diseases, since patients are generally hospitalized throughout treatment, higher drug toxicity levels and costs may be acceptable, and IV dosing would be preferred. A key factor in effectiveness for a drug for this indication is penetration of the blood-brain barrier, as all three parasites infect the CNS. Obatoclax has previously been shown to have high CNS levels [29], and has been in phase II clinical trials in IV formulation. If sufficient CNS levels are not reached, intrathecal delivery, which has proven effective for some antibiotics [65] could also be considered. It should be noted that due to the extreme rarity of CNS and systemic diseases caused by the free-living ameba, clinical trials would not be feasible. Demonstration of efficacy in animal models and an established record of safety in previous human trials can lead to approval for clinical use in these cases. For ocular disease, multiple animal models exist, which could be used to test safety and efficacy [66].

Current drug treatment for *Trypanosoma brucei* is complicated, often requiring repeated parenteral administration of drugs or drug combinations under clinical supervision to treat either Stage 1 (hemolymphatic) or Stage 2 (CNS-infiltrated) disease. Fortunately, one new drug, fexinidazole, was approved in 2018 as the first all-oral treatment of both stage 1 and 2 West African (gambiense) trypanosomiasis, and clinical trials are ongoing regarding its potential to treat East African (rhodesiense) disease [52]. Although, this is excellent progress, the goal should be to generate a portfolio of drugs in the case (as has happened for trypanocidal drugs in the past) of resistance emerging. Of the four drugs tested here against *T. brucei*, all except nithiamide yielded $EC_{50}$ values less than the current Stage 1 treatment option, pentamidine ($EC_{50}$ = 0.11 µM), including, encouragingly, the brain-penetrant obatoclax ($EC_{50}$ = 0.044 µM). Further investigations, including with other strains of *T. brucei*, are ongoing.

In this study we have identified compounds efficacious against a diverse collection of parasites. Such broad-spectrum drugs could be of great utility in situations where diagnosis is uncertain (for instance, in the case of the free-living ameba which can all cause encephalitis) or when a patient is infected with multiple pathogens. Additionally, there could be economic advantages to a "one drug multiple-bug" approach. Currently, development of drugs for neglected tropical diseases is stymied by the high cost and low economic incentive to provide drugs for developing countries [67]. A single drug with broad spectrum activity would have a larger potential market, altering the cost benefit analysis for pharmaceutical companies. With further characterization of *in vivo* efficacy and toxicity, the compounds identified in this study, in particular obatoclax and anisomycin, have the potential to become useful tools in the treatment of multiple parasitic disease.

## Methods

### Culture and strains used

*Entamoeba histolytica* strain HM-1:IMSS trophozoites were grown and maintained at 37˚C in TYI media under standard conditions [68].

*Giardia lamblia* strain WB was grown and maintained at 37˚C in TYI media.

*Acanthamoeba castellanii* strain Ma trophozoites (T4 genotype) were cultured at 28˚C in PYG medium according to a modified technique [69].

*Naegleria fowleri* strain KUL trophozoites were axenically cultured in Nelson's medium supplemented with 10% FBS at 37˚C [70, 71].

*Balamuthia mandrillaris* ATCC strain PRA-291 were grown in axenic modified Cerva's medium [72] at 37˚C and 5% $CO_2$. Encystation was induced by incubation in medium with galactose (12% final concentration) as previously described [21, 73].

*Cryptosporidium parvum* strain Iowa oocysts were obtained from Bunch Grass Farm (Deary, ID) [58].

*Plasmodium falciparum* strain Dd2attB (MRA-843) were obtained from MR4. Parasites were grown in human erythrocytes (2% hematocrit) in RPMI 1640 media supplemented with 0.25% Albumax II (GIBCO Life Technologies), 2 g/L sodium bicarbonate, 0.1mM hypoxanthine, 25mM HEPES (pH 7.4), and 50μg/L gentamycin, at 37˚C, 5% $O_2$, and 5% $CO_2$ [74].

Juvenile *Schistosoma mansoni* from the Biomedical Research Institute were obtained from infected mice ~3 weeks post-infection by hepatic portal vein perfusion using 37˚C DMEM with 5% heat inactivated FBS. Worms were rinsed to remove mouse blood and cultured at 37˚C/5% $CO_2$ in Basch Medium 169 supplemented with 1X antibiotic-antimycotic [75].

Bloodstream forms of *T. b. brucei* Lister 427 were cultured *in vitro* in T25 vented flasks (Thermo Fisher Scientific) in a humidified atmosphere of 5% $CO_2$ at 37˚C, using a modified Iscove's medium (HMI-11) with 20% heat-inactivated fetal bovine serum (FBS; Gibco, Carlsbad, CA) [76]. Parasites are maintained in log-phase growth (between $1\times10^5$ and $1\times10^6$ parasites/mL) and passaged every 48 h.

hFF-1 cells (gift from John Boothroyd lab, Stanford University) were cultured in DMEM with 1g/L glucose, Sodium Pyruvate and L-glutamine (Gibco) completed with 10% FBS, and 100 U/mL penicillin/streptomycin at 37˚C with 5% $CO_2$.

### hFF-1 cytotoxicity experiments

hFF-1 cells were plated in 96-well plates in 50μl DMEM at a density of 2000 cells/well, and incubated at 37˚C for ~6h to allow adherence. Drug was then added in 50μl volume to give final concentrations from 60–0.03μM, and plates were returned to the incubator. After 72h, viability was assed using CellTiter-Glo Luminescent Cell Viability Assay (Promega). Results are from two independent experiments and $EC_{50}$s were calculated using the GraphPad prism.

### Speed of killing experiments

**E. histolytica.**   To determine the kinetics of our compounds activity against *E. histolytica*, we adapted the viability assay we developed in our previous studies [6] and performed a time course to measure changes in the number of live trophozoites over 48 hours. *E. histolytica* trophozoites (10,000 parasites in 150μl media) were seeded into 96-well plates and allowed to grow overnight in an anaerobic chamber. The next day, a single plate was removed for analysis, and drug was added to remaining plates at 2x the previously calculated $EC_{50}$ concentration. Parasite viability was assayed at 10, 24 and 48h using the live cell marker fluorescein diacetate (Sigma) as previously described [6]. Three independent biological replicates were performed.

**B. mandrillaris.**   For the speed of killing assay, trophozoites of *B. mandrillaris* (10,000 parasites in 100 μL) were treated with the compounds in black, clear bottom 96-well plates for 8, 24, and 48h. Viability was determined after incubation with fluorescein diacetate for 30 minutes, fixing the trophozoites with 4% PFA and measuring fluorescence using a Tecan Infinite M1000 pro fluorometer following incubation. The percentage viability of trophozoites treated with different compounds at different time points was calculated. 0.5% DMSO was used as a negative control and 15.5μM nitroxoline (2x the published $EC_{50}$ [21]) was used as a positive control. Three independent biological replicates were performed.

***Balamuthia* encystment induction.**   *Balamuthia* trophozoites (50,000 in 500μL media per well) were plated in in 24 well plates and incubated at 37˚C for 72h with drug (15.6 μM nitroxoline and 3.2μM obatoclax) or DMSO (0.5%). Trophozoites were fixed by adding 8% paraformaldehyde to a final concentration of 4%, and cysts were counted using a hemocytometer. Trophozoites and cysts were distinguished by visual inspection under light microscopy. Results are expressed as number of cysts per mL of media.

***Balamuthia* Recrudescence.**   Recrudescence assays to asses recovery from drug treatment were adapted from a previous study [21]. *B. mandrillaris* trophozoites ($10^5$ cells in 4mL of media) were exposed to obatoclax at a concentration of 3.2μM for 72h at 37˚C. Treated *B. mandrillaris* parasites were then centrifuged, washed once with PBS and resuspended in 4mL of complete DMEM media. Resuspended trophozoites (0.5mL) were transferred in duplicate to each well of a 24-well plate containing hFF-1 cells that had been seeded at $5x10^4$ per well and incubated for 72h at 37˚C. Plates were fixed with 4% PFA after 3, 5, 7 and 9 days, and the percent of the HFF monolayer lysed was observed using light microscopy and recorded for each timepoint. 0.5% DMSO was used as a negative control and 15.6 μM nitroxoline [21] was used as a positive control. Three independent biological replicates were performed.

## Parasite drug susceptibility testing

***Giardia lamblia*.**   The compounds were screened against *G. lamblia* trophozoites following an ATP-bioluminescence based assay for cell growth and survival [77]. For primary screen, parasites were seeded into a 96-well plate at 5,000 trophozoites per well with a final concentration of test compound of 50μM in 100μl media. Negative controls in the screen plates contained 0.5% DMSO and positive controls contained 50μM of metronidazole. For 16-point dose response study, stock compounds were serially diluted with DMSO to yield a final 16-point concentration range spanning 0.0015μM-50μM. Assay plates were incubated for 48h at 37˚C in the GasPak EZ gas-generating anaerobe pouch system (VWR), and at the end of incubation 50μL of CellTiter-Glo Luminescent Cell Viability Assay (Promega) was added in each well of the 96-well plates to induce cell lysis. The resulting ATP-bioluminescence of the trophozoites was measured at room temperature using an Envision plate reader from PerkinElmer. Percent inhibition was calculated, and the relative dose response data in triplicate were exported to GraphPad Prism software 8.0 for $EC_{50}$ calculations and statistical analysis.

***Acanthamoeba castellanii* and *Naegleria fowleri*.**   Trophozoites were screened using the same protocol as *Giardia*, with the following modifications: no GasPak was used; for *N. fowleri*, 10,000 trophozoites were seeded per well and 50μM amphotericin B was used as a positive control; for *A. castellanii* 5,000 trophozoites were seeded per well and 50μM of chlorhexidine was used as a positive control [70].

***Balamuthia mandrillaris*.**   Trophozoites and cysts were screened using protocols adapted from [21]. Briefly, 6000 trophozoites or 4000 cysts were seeded into opaque 96-well plates and incubated with drug (50–0.4μM) or DMSO (0.5%) in 100μl media for 72h. Viability was assessed using CellTiter-Glo. $EC_{50}$ values were determined using the GraphPad Prism

4-parametric sigmoidal curve-fitting model, with the bottom and top constraints set to 0 and 1, respectively.

***Plasmodium falciparum.*** Assays were performed as in [78]. Cultures were grown in 125μL volume in 96-well plates containing serial dilution of drugs in triplicate. Growth was initiated with ring-stage parasites at 1% parasitemia and 1% hematocrit. Drug (100–0.1μM) or vehicle (DMSO) was added and plates were incubated for 72h. Growth was terminated by fixation with 1% formaldehyde and parasitized cells were stained with 50nM YOYO-1 (Invitrogen). Parasitemia was determined by flow cytometry. Data were analyzed by BD C6 Accuri C-Sampler software, and $EC_{50}$ curves plotted by GraphPad Prism.

***Cryptosporidium parvum.*** Growth inhibition was determined by an Immunofluorescence assay adapted from [79]. Oocysts were excysted by treatment with 10mM hydrochloric acid (10 min at 37˚C), followed by exposure to 2mM sodium taurocholate (Sigma-Aldrich) in PBS for 10 min at 16˚C. Excysted oocysts were then added to >95% confluent HCT-8 cell monolayers in 384-well clear-bottom plates at a concentration of 5,500 Iowa isolate oocysts per well. Compounds were added just before or 3h after infection, and assay plates were incubated for 48h post-infection at 37˚C under 5% $CO_2$. Wells were then washed three times with PBS containing 111mM D-galactose, fixed with 4% formaldehyde in PBS for 15 min at room temperature, permeabilized with 0.25% Triton X-100 for 10 min at 37˚C, washed three times with PBS with 0.1% Tween 20, and blocked with 4% bovine serum albumin (BSA) in PBS for 2 h at 37˚C or 4˚C overnight. Parasitophorous vacuoles were stained with 1.33 micrograms/ml of fluorescein-labeled *Vicia villosa* lectin (Vector Laboratories) diluted in 1% BSA in PBS with 0.1% Tween 20 for 1h at 37˚C, followed by the addition of Hoechst 33258 (AnaSpec) at a final concentration of 0.09mM for another 15 min at 37˚C. Wells were then washed five times with PBS containing 0.1% Tween 20. A Nikon Eclipse TE2000 epifluorescence microscope with an automated stage was programmed using NIS-Elements Advanced Research software (Nikon, USA) to focus on the center of each well and take a 3-by-3 composite image using an EXi Blue digital camera (QImaging, Canada) with a 20x objective (numerical aperture, 0.45). Nucleus and parasite images were analyzed using macros developed on the ImageJ platform (National Institutes of Health). The only modification from the published macro used to count parasites was that the lower size threshold for parasites was decreased from 16.5 to 4 pixels (1 pixel = 0.65 micrometers). Graphs were plotted and $EC_{50}$ values calculated using GraphPad Prism software, version 6.01.

***Schistosoma mansoni.*** Juvenile worms were randomly distributed to 24-well plate. To each well was added 2mL drug, diluted to the desired concentration (0.05–5μM) or fresh medium. After 72-hour drug treatment, the mobility of worms was examined. Worms that were not mobile within the 2-minute observation time period were counted as dead worms. After 72h, drug-containing media was removed worms were washed once in medium and returned to fresh-drug free media. Worms were observed for 3 days to monitor recovery from drug treatment. For $EC_{50}$ calculations, two independent biological experiments were performed. For imaging, drug treated worms were imaged on a Zeiss Stemi 508 microscope at 5x magnification. Untreated control worms were washed briefly in 0.6M $MgCl_2$ and then in PBS + Triton X-100 (0.3%) prior to imaging to reduce motility. To avoid morphological changes of worms, fixation step was omitted for both drug-treated and control worms.

***Trypanosoma brucei.*** The SYBR Green assay ([80] as modified by Monti *et al.* [81]) was employed to measure the effect of test compounds on cell viability. Compounds in 100% DMSO stocks were diluted in 100μL HMI-10 modified medium in 96-well plates (Corning 3903) such that the final DMSO concentration was 0.5%. Eight-point dose response assays (ranging from final concentrations of compound from 4μM to 0.8nM) were set up in duplicate. Bloodstream-form trypanosomes in log-phase growth were then diluted to $2x10^5$ cells/

mL in HMI-10 medium and dispensed into the previously prepared 96-well plates at 100μL per well. After 72h, trypanosomes were lysed by the addition of 50μL/well of lysis buffer containing $1 \times$ SYBR Green. Plates were read on the 2104 EnVision multilabel plate reader (PerkinElmer) with excitation at 485 nm/emission at 535 nm. The activity of test compounds was normalized against controls from the same plate according to the following formula: Activity (%) = [1 – (FCpd – blank) / (FNeg – blank)] × 100, where FCpd corresponds to the emitted fluorescent signal expressed in arbitrary fluorescence units for the test compound; and FNeg and blank correspond to the mean fluorescent signal of the negative control wells and the background signal, respectively. $EC_{50}$ values, *i.e.*, the concentration of drug required to inhibit trypanosome growth by 50%, were calculated using GraphPad Prism software, version 6.00 for Apple Macintosh. Each assay was performed as three experimental replicates and means ±SD values are shown.

**Animal studies with *Cryptosporidium* and drug susceptibility.** *In vivo* mouse experiments were performed following a protocol based on [58]. All NOD SCID gamma mouse studies were performed in compliance with animal care guidelines and were approved by the University of Vermont Institutional Animal Care and Use Committee. NOD SCID gamma mice with normal flora (NOD.Cg-Prkdcscid Il2rgtm1Wjl/SzJ) were purchased from The Jackson Laboratory (Bar Harbor, ME, USA) and were housed for at least a week for acclimatization. At the age of 4 to 5 weeks, mice were infected with $10^5$ *C. parvum* Iowa isolate oocysts. Fecal oocyst shedding is detected 6 days after infection using a quantitative PCR (qPCR assay), so treatment was started on day 7 after infection. Mice (4 per experimental group) were treated orally with prodigiosin at 25mg/kg twice daily, with MMV665917 at 60mg/kg twice daily as a positive control [58], or with vehicle alone. A final concentration of 5% DMSO was used in 100μl of 1% HPMC per dose. Mice were treated on days 7, 8, 9, and 10 post-infection. Oocyst shedding in feces was monitored by qPCR just prior to treatment and 1 day following completion of treatment.

## Compound Sources

Compounds were obtained from the following vendors: Prodigiosin: BOC Sciences (Catalog number 82-89-3); Obatoclax: Apex Bio (Catalog number A419); Anisomycin: Sigma (Catalog number A9789); Nithiamide: Sigma (Catalog number 33978); Venetoclax: Selleckchem (Catalog Number S8048); Navitoclax: Selleckchem (Catalog Number S1001); A-1331852: Selleckchem (Catalog Number S7801); A-1210477: Selleckchem (Catalog Number S7790); S63845: Selleckchem (Catalog Number S8383).

## Supporting information

**S1 Table. Potency of BCL-2 inhibitors against the parasite systems tested.** The $EC_{50}$s for BCL-2 inhibitors for each parasite tested are shown. All concentrations are micromolar. (XLSX)

## Acknowledgments

We would like the thank the Stanford HTBC for technical help and members of the Singh lab for valuable discussion. Important consultation was provided by SPARK advisors, especially Toni Kline, Steve Schow, Marcus Parrish, Jeewon Kim, and Kevin Grimes.

## Author Contributions

**Conceptualization:** Gretchen Ehrenkaufer.

**Data curation:** Gretchen Ehrenkaufer.

**Funding acquisition:** Gretchen Ehrenkaufer, Anjan Debnath, Conor R. Caffrey, Christopher D. Huston, Upinder Singh.

**Investigation:** Gretchen Ehrenkaufer, Pengyang Li, Erin E. Stebbins, Monica M. Kangussu-Marcolino, Anjan Debnath, Corin V. White, Matthew S. Moser, Jolyn Gisselberg, Steven C. Wang, Ana Hervella Company, Ludovica Monti, Conor R. Caffrey, Christopher D. Huston.

**Methodology:** Gretchen Ehrenkaufer, Pengyang Li, Monica M. Kangussu-Marcolino, Anjan Debnath, Corin V. White, Matthew S. Moser, Jolyn Gisselberg.

**Project administration:** Gretchen Ehrenkaufer, Joseph DeRisi, Ellen Yeh, Upinder Singh.

**Resources:** Bo Wang.

**Supervision:** Joseph DeRisi, Ellen Yeh, Conor R. Caffrey, Christopher D. Huston, Bo Wang, Upinder Singh.

**Writing – original draft:** Gretchen Ehrenkaufer, Monica M. Kangussu-Marcolino, Anjan Debnath, Corin V. White.

**Writing – review & editing:** Gretchen Ehrenkaufer, Christopher D. Huston.

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
