## [Decision Letter · Decision Letter 0]

22 Dec 2019

Dear Dr. Singh:

Thank you very much for submitting your manuscript "Identification of anisomycin, prodigiosin and obatoclax as compounds with broad-spectrum anti-parasitic activity" (PNTD-D-19-01940) for review by PLOS Neglected Tropical Diseases. Your manuscript was fully evaluated at the editorial level and by independent peer reviewers. The reviewers appreciated the attention to an important topic but identified some aspects of the manuscript that should be improved.

We therefore ask you to modify the manuscript according to the review recommendations before we can consider your manuscript for acceptance. Your revisions should address the specific points made by each reviewer.

(1) A letter containing a detailed list of your responses to the review comments and a description of the changes you have made in the manuscript.

(2) Two versions of the manuscript: one with either highlights or tracked changes denoting where the text has been changed (uploaded as a "Revised Article with Changes Highlighted" file ); the other a clean version (uploaded as the article file).

(3) If available, a striking still image (a new image if one is available or an existing one from within your manuscript). If your manuscript is accepted for publication, this image may be featured on our website. Images should ideally be high resolution, eye-catching, single panel images; where one is available, please use 'add file' at the time of resubmission and select 'striking image' as the file type. 

Please provide a short caption, including credits, uploaded as a separate "Other" file. If your image is from someone other than yourself, please ensure that the artist has read and agreed to the terms and conditions of the Creative Commons Attribution License at http://journals.plos.org/plosntds/s/content-license (NOTE: we cannot publish copyrighted images). 

(4) Appropriate Figure Files 

Please remove all name and figure # text from your figure files upon submitting your revision. Please also take this time to check that your figures are of high resolution, which will improve both the editorial review process and help expedite your manuscript's publication should it be accepted. Please note that figures must have been originally created at 300dpi or higher. Do not manually increase the resolution of your files. For instructions on how to properly obtain high quality images, please review our Figure Guidelines, with examples at: http://journals.plos.org/plosntds/s/figures

While revising your submission, please upload your figure files to the Preflight Analysis and Conversion Engine (PACE) digital diagnostic tool, https://pacev2.apexcovantage.com/ PACE helps ensure that figures meet PLOS requirements. To use PACE, you must first register as a user. Then, login and navigate to the UPLOAD tab, where you will find detailed instructions on how to use the tool. If you encounter any issues or have any questions when using PACE, please email us at figures@plos.org.

We hope to receive your revised manuscript by Feb 20 2020 11:59PM. If you anticipate any delay in its return, we ask that you let us know the expected resubmission date by replying to this email.

To submit your revised files, please log in to https://www.editorialmanager.com/pntd/

Sincerely,

Alvaro Acosta-Serrano

Deputy Editor

Alvaro Acosta-Serrano

Deputy Editor

Reviewer's Responses to Questions

**Key Review Criteria Required for Acceptance?**

**Methods**

-Are the objectives of the study clearly articulated with a clear testable hypothesis stated?

-Is the study design appropriate to address the stated objectives?

-Is the population clearly described and appropriate for the hypothesis being tested?

-Is the sample size sufficient to ensure adequate power to address the hypothesis being tested?

-Were correct statistical analysis used to support conclusions?

-Are there concerns about ethical or regulatory requirements being met?

Reviewer #1: Yes to all. The goal here was to test four repurposed drugs that are effective versus Entamoeba in vitro versus other pathogenic protists. In each case the investigators, which are experts in each organism tested and from numerous universities, use assays that allow them to determine the EC50 for each compound.

Reviewer #2: A previous screen of the ReFrame compound library against Entamoeba histolytica led to the discovery of 4 compounds, Prodigiosin, Obatoclax, Anisomycin, and Nithiamide, that have previously been in human trials or use. The authors report on advanced testing of these 4 compounds on advanced tests for E. histolytica bioactivity and to investigate a panel of other parasites including pathogenic free-living amoeba, cryptosporidium, plasmodium, and Schistosoma mansoni. The study design is appropriate to shed light on the broad activity of some of these compounds. I had no concerns about the methods

**Results**

-Does the analysis presented match the analysis plan?

-Are the results clearly and completely presented?

-Are the figures (Tables, Images) of sufficient quality for clarity?

Reviewer #1: Yes to all. The results are clearly presented in tables, figures, and text. While the results vary from drug to drug and organism to organism, enough of them are promising to make the study of interest and enough results are negative to assure that investigators were honest.

Reviewer #2: The results are clearly presented and well written. The results show they have much promise, in vitro, for rapid action against E. histolytica, though somewhat disappointing in the cidal washout experiment. Some of the drugs have activity against other of the panel of parasites tested, though the only in vivo study was a negative study with Prodigiosin in C. parvum infection of NOD/SCID/gammaKO mice. Despite this negative study, there is abundant information to justify rapid publication of the results. 

 The figures are of good quality, except in the figure legend 4, it states we should see dots representing the values of the 4 animals tested but I can only see a very few dots. 

On Page 22 of the PDF, line 309, there is a reference to Table S1, but the Table they are referring to in line 309 should actually be Table 1. Furthermore Table S1 should be moved down towards the end of the paper where it is discussed.

**Conclusions**

-Are the conclusions supported by the data presented?

-Are the limitations of analysis clearly described?

-Do the authors discuss how these data can be helpful to advance our understanding of the topic under study?

-Is public health relevance addressed?

Reviewer #1: yes to all. Their general hypothesis that there are repurposable drugs out there that might be tested versus pathogenic protists was confirmed. With the exception of one negative mouse study of Cryptosporidium, none of the drugs were tested in animal models, and no mechanism was investigated. Despite these limitations, the study has impact based upon the breadth of organisms studied.

Reviewer #2: The conclusions are reasonably well supported by the data but could be improved by my suggestions below. The relevance to advancing the field and therapies are discussed as well as (indirectly) the public health relevance. 

1) I feel the paper's findings' applicability and promise would be way easier to evaluate if the authors made a comparative table of the 4 drugs, especially including their pharmacokinetic (PK) parameters. The PK should include, where ever available: plasma peak (Cmax) and exposure (AUC) information after various doses when give by oral (and IV) forms. This could be only the human data or better yet, include major efficacy species, such as mice. The plasma protein binding of each drug would be useful information as well as oral bioavailability and CNS penetration. Some of this data is (nonquantitatively) mentioned in passing, and is probably available in the public domain. Then the discussion could be expanded into whether higher or lower doses that have previously been given to humans would likely be necessary, or even whether intrathecal delivery for the amoebic encephalitis cases might be necessary. 

2) The above data might further inform the negative study on C. parvum and expand the discussion here. For instance is prodigiosin so well absorbed that it might not be delivered to the terminal ileum/proximal colon where the C. parvum infection is highest.

3) The host cell toxicity data is only shown for one mammalian cell line, HepG2, and is certainly alarming for Anisomycin, Prodigiosin, and Obatoclax where the CC50 values for HepG2 are well below what is considered efficacious for many parasites. The authors don't discuss this data, but I feel they should. Perhaps that data is mitigated by the sheer fact that these therapies have been in humans already, but it's all relative to the levels achieved in vivo and the serum protein binding of the compounds (see #1 above).

**Editorial and Data Presentation Modifications?**

Reviewer #1: in the abstract line 54 and discussion line 424 they should be careful to say that there are no treatments for brain infection with Acanthamoeba, although there are available topical treatments for eye infections.

Reviewer #2: See above, minor revisions

**Summary and General Comments**

Reviewer #1: well done study by a large group of experts.

Reviewer #2: Outstanding paper and only minor revisions are needed

PLOS authors have the option to publish the peer review history of their article (what does this mean?). If published, this will include your full peer review and any attached files.

Reviewer #1: Yes: John Samuelson

Reviewer #2: Yes: Wes Van Voorhis

---

## [Editor Report · Decision Letter 1]

18 Feb 2020

Dear Dr. Singh,

We are pleased to inform you that your manuscript 'Identification of anisomycin, prodigiosin and obatoclax as compounds with broad-spectrum anti-parasitic activity' has been provisionally accepted for publication in PLOS Neglected Tropical Diseases.

Before your manuscript can be formally accepted you will need to complete some formatting changes, which you will receive in a follow up email. A member of our team will be in touch within two working days with a set of requests.

Best regards,

Alvaro Acosta-Serrano

Deputy Editor

---

## [Editor Report · Acceptance letter]

13 Mar 2020

Dear Dr. Singh,

We are delighted to inform you that your manuscript, "Identification of anisomycin, prodigiosin and obatoclax as compounds with broad-spectrum anti-parasitic activity," has been formally accepted for publication in PLOS Neglected Tropical Diseases.

Best regards,

Serap Aksoy

Editor-in-Chief

Shaden Kamhawi

Editor-in-Chief
